# The Mineralocorticoid Receptor on Smooth Muscle Cells Promotes Tacrolimus-Induced Renal Injury in Mice

**DOI:** 10.3390/pharmaceutics15051373

**Published:** 2023-04-29

**Authors:** Stefanny M. Figueroa, Jean-Philippe Bertocchio, Toshifumi Nakamura, Soumaya El-Moghrabi, Frédéric Jaisser, Cristián A. Amador

**Affiliations:** 1Institute of Biomedical Sciences, Universidad Autónoma de Chile, Santiago 8910060, Chile; 2INSERM UMRS1138, Sorbonne Université, Université de Paris, Centre de Recherche des Cordeliers, 75006 Paris, France; 3Faculty of Medicine and Science, Universidad San Sebastián, Santiago 7510156, Chile

**Keywords:** tacrolimus, nephrotoxicity, mineralocorticoid receptors

## Abstract

Tacrolimus (Tac) is a calcineurin inhibitor commonly used as an immunosuppressor after solid organ transplantation. However, Tac may induce hypertension, nephrotoxicity, and an increase in aldosterone levels. The activation of the mineralocorticoid receptor (MR) is related to the proinflammatory status at the renal level. It modulates the vasoactive response as they are expressed on vascular smooth muscle cells (SMC). In this study, we investigated whether MR is involved in the renal damage generated by Tac and if the MR expressed in SMC is involved. Littermate control mice and mice with targeted deletion of the MR in SMC (SMC-MR-KO) were administered Tac (10 mg/Kg/d) for 10 days. Tac increased the blood pressure, plasma creatinine, expression of the renal induction of the interleukin (IL)-6 mRNA, and expression of neutrophil gelatinase-associated lipocalin (NGAL) protein, a marker of tubular damage (*p* < 0.05). Our study revealed that co-administration of spironolactone, an MR antagonist, or the absence of MR in SMC-MR-KO mice mitigated most of the unwanted effects of Tac. These results enhance our understanding of the involvement of MR in SMC during the adverse reactions of Tac treatment. Our findings provided an opportunity to design future studies considering the MR antagonism in transplanted subjects.

## 1. Introduction

Calcineurin is a Ca^2+^-calmodulin-dependent phosphatase that is activated after T-cell receptor agonism [1]. Calcineurin causes dephosphorylation of the nuclear factor of activated T cells. This favors its nuclear translocation and further transcription of different cytokines, including interleukin (IL)-2, IL-4, IL-17, interferon (IFN)-γ, and tumor nuclear factor (TNF)-α [2]. In contrast, calcineurin inhibitors (CIs) hinder the phosphatase activity of calcineurin by forming complexes with cytosolic immunophilins [3,4]. The major CIs are cyclosporin-A (CsA) and tacrolimus (Tac), both widely used as a preventive treatment for allograft rejection of solid organs, particularly kidney transplants [4,5,6].

Although CIs can efficiently decrease the incidence of allograft rejection when used along with other immunosuppressive agents [4,7], they might cause acute and/or chronic renal injury [8]. Tac administration induces an increase in blood pressure and plasma creatinine in post-renal transplant patients [9,10,11]. Although the precise mechanisms of Tac-induced nephrotoxicity are not entirely comprehended, studies on murine models have revealed that it is linked to acute hemodynamic impairment [12,13]; reduced natriuresis [14]; elevated proinflammatory lymphocytes [15]; heightened levels of proinflammatory cytokines; such as IL-6 [14,15]; and the induction of a fibrotic phenotype [16,17].

Some studies have proposed mineralocorticoid receptor (MR) antagonism as an additional pharmacological strategy for preventing renal rejection because it has beneficial effects on patients with cardiovascular and renal diseases [18,19,20]. While the mechanisms by which MR antagonists (MRAs) provide renal and cardiovascular protection are not fully elucidated, the beneficial effects of MRAs are probably linked to the regulation of the expression and/or activity of ion channels (such as Na^+^, K^+^, and Ca^2+^ ion channels), reduction in oxidative stress, and decrease in fibrosis, extracellular matrix remodeling, and the inflammatory state [21,22]. The use of MRAs, such as spironolactone (Spiro) and/or eplerenone, concomitantly with CIs, can reduce the morbidity and mortality in kidney transplant patients [20,23], prevent the decrease in the glomerular filtration rate (GFR), and reduce proteinuria and structural damage [24,25,26], suggesting that MRAs may prevent the damage caused by CIs. Additionally, experiments on animal models have shown that MRAs can reduce or prevent the adverse effects generated by the use of CIs [27,28,29], such as renal remodeling (apoptosis and fibrosis) and the regulation of vasoactive factors resulting from CsA treatment [21].

Since MR is expressed in various cell types, recent research has investigated its role in different cell types. For instance, the inhibition of MR in smooth muscle cells (SMC), but not in endothelial cells, was discovered to be responsible for mitigating CsA-induced nephrotoxicity in mice [27]. Here, we assessed whether Spiro prevents Tac-induced nephrotoxicity and whether SMC-MR is also involved.

## 2. Materials and Methods

### 2.1. Animals

Wild-type (WT) and SMC-MR-KO C57BL/6 mice were used in this study. Genetic ablation of the MR in SMC was achieved by mating transgenic mice expressing Cre recombinase in SMC (SMC-Cre-Er^T2^ mice) with MR-flox mice, which resulted in SMC-MR-KO mice [30,31]. Efficient ablation of MR from SMC was corroborated for all mice. Littermate MR-Flox mice were used as controls. All animal studies were conducted following the National Institutes of Health Guide and European Community directives for the Care and Use of Laboratory Animals (European Directive, 2010/63/UE), approved by the local animal ethics committee (APAFIS#7304–2016102013052714 v2), and conducted according to the INSERM animal care and use committee guidelines.

### 2.2. Experimental Protocol

Eight-week-old C57BL/6 male mice were administered a low-salt diet (0.01% NaCl) for seven days to sensitize the renin–angiotensin–aldosterone system, as previously described [32]. The mice were then treated with a vehicle solution (EtOH 95% and Cremophor Sigma, in a ratio of 1:4 (*v*/*v*)), Tac (1 or 10 mg/Kg per day diluted in the vehicle solution, volume ~150 μL) subcutaneously for 10 days, or Spiro (100 mg/Kg per day in methylcellulose) per gavage. Throughout the study, all animals were kept in an environment with a 12 h light/dark cycle, 50% humidity, and a temperature of 21 °C. They were provided with proper ventilation and given ad libitum access to food and water.

### 2.3. Blood Pressure Measurements

Systolic blood pressure was measured in conscious mice via tail cuff plethysmography. All mice were trained for three days before the measurements were taken. Their blood pressure was assessed at −6, 0, 2, 6, and 10 days of Tac treatment. Consecutive pulse readings (8–10) were recorded for each mouse at each measurement using Visitech BP-2000, as previously reported [33].

### 2.4. Biochemical Assays

Following euthanasia, blood was drawn from the mice and centrifuged to obtain plasma, which was stored at a temperature of −20 °C. Plasma creatinine was analyzed by an enzymatic method using Konelab v.7.0.1 automate (Pierce/Thermo Fischer Scientific, Rockford, IL, USA).

### 2.5. Western Blotting

The kidneys of the mice were collected, and proteins were extracted with a sodium dodecyl sulfate (SDS) 1% buffer (pH = 7.4). Then, 30 μg of protein was separated by SDS-polyacrylamide gel electrophoresis (Bio-Rad, Hercules, CA, USA), blotted onto nitrocellulose membranes (Amersham ECL Plus; GE Healthcare Life Sciences, Freiburg, Germany), and probed with primary antibodies, including anti-neutrophil gelatinase-associated lipocalin (NGAL, #ab63929) and β-actin (#ab8227) (both antibodies were purchased from Abcam, Cambridge, UK). Subsequently, anti-rabbit HRP secondary antibodies (GE Healthcare Life Sciences) were utilized, and specific binding was detected by enhanced chemiluminescence (Amersham ECL Plus; GE Healthcare Life Sciences). The resulting images were analyzed using a Fujifilm Luminescent Image Analyzer LAS4000 System (Tokyo, Japan), and densitometry analysis was conducted using ImageJ 1.53a software (US National Institutes of Health, Bethesda, MD, USA) to quantify the images of the blots.

### 2.6. Real-Time PCR

The kidneys were collected, and total RNA was extracted using the TRIzol Reagent. Next, cDNA was synthesized from 2 μg of RNA using the Superscript II Reverse Transcriptase Kit (all from Life Technologies Corporation, Carlsbad, CA, USA). Real-time PCR reactions were performed using a Bio-Rad Thermal Cycler (Cergy-St-Christophe, France) (iCycler iQ apparatus) and the transcript levels were estimated by the SYBR Green method. The sequences of the mouse primer pairs were as follows: Ubiquitin-C (UBC), (F) 5′-CGGAGTCGCCCGAGGTCACA-3′, (R) 5′-GGGCTCGACCTCCAGGGTGAT-3′; IL-6, (F) 5′-CTCTGGGAAATCGTGGAAATG-3′, and (R) 5′-AAGTGCATCATCGTTGTTCATACA-3′. All PCR products were analyzed by the melting-curve program to confirm the specificity of amplification. The results were analyzed using the standard curve method, and the abundance of mRNA was calculated in relation to the amount of UBC for each sample.

### 2.7. Statistics

The data are presented as mean ± SEM. Depending on the need, one-way or two-way analysis of variance was performed, followed by Tukey’s post hoc test for data analysis. All analyses were performed using GraphPad Prism V9.3.1 (GraphPad Software, San Diego, CA, USA), and all differences among and between groups were considered to be statistically significant at *p* < 0.05.

## 3. Results

### 3.1. A High Dose of Tac Promotes Renal Damage in Mice

A dose of 10 mg/Kg/d Tac (Tac-10) induced an increase in the systolic blood pressure (140.02 ± 12.74 vs. 112.25 ± 9.10 mmHg in control) and promoted a rise in plasma creatinine in WT mice, which was significant after treatment for 10 days (Figure 1A,B, respectively), while 1 mg/Kg/d Tac (Tac-1) had no significant effect. Similarly, only Tac-10 exhibited the ability to induce a surge in IL-6 mRNA at the renal level (*p* < 0.001) (Figure 1C). It raised the expression of NGAL protein (in its non-glycosylated ~22-kDa precursor and glycosylated ~25-kDa mature form), which serves as a tubular damage marker, by twofold as compared to its basal levels (*p* < 0.05) (Figure 1D). Since Tac-10 increased the blood pressure associated with renal damage after 10 days without generating toxicity or altering in a significant way the body weight in mice (data not shown), further experiments were performed using this dose.

### 3.2. MR Antagonism Prevents Renal Damage Induced by Tac-10

The co-administration of Spiro (100 mg/Kg/day) during Tac-10 treatment in WT mice prevented the increase in systolic blood pressure after 10 days (*p* < 0.01) (Figure 2A). Likewise, Spiro administration prevented the elevation of plasma creatinine levels and renal expression of IL-6 mRNA induced by Tac-10 (*p* < 0.01), as depicted in Figure 2B,C, respectively. The overexpression of NGAL in the kidneys of WT mice induced by Tac-10 was inhibited by Spiro (*p* < 0.05) (Figure 2D). These results suggested that Tac-10 promotes an increase in blood pressure and kidney injury, at least partly, through the activation of MR.

### 3.3. Tac-10 Promotes Renal Injury through the MR Expressed in the SMC

In a separate investigation, we demonstrated that the SMC-MR significantly contributes to CsA-induced nephrotoxicity and alterations in renal hemodynamics in mice [27]. Based on those findings, we examined whether the SMC-MR was also involved in the kidney injury induced by Tac. To determine this, we used the same strategy of SMC-specific deletion of MR (SMC-MR-KO) [30] to assess the kidney injury induced by Tac-10 in mice (Figure 3A).

The SMC-MR-KO mice showed no significant alteration in systolic blood pressure under Tac-10 treatment compared to the systolic blood pressure of the MR-Flox control mice (Figure 3B). Tac-10 was found to significantly elevate diastolic blood pressure in MR Flox mice (data not shown). In addition, plasma creatinine increased significantly after 10 days of Tac-10 treatment in MR-Flox mice (13.54 ± 4.94 vs. 6.38 ± 1.49 in control) but not in SMC-MR-KO mice (Figure 3C). The renal IL-6 mRNA levels increased significantly after treatment with Tac-10 in MR-Flox mice (*p* < 0.05) but not in SMC-MR-KO mice (Figure 3D). Finally, a significant renal induction of NGAL protein was recorded after Tac-10 treatment in MR-Flox mice but not in SMC-MR-KO mice (*p* < 0.01) (Figure 3E). Thus, the deletion of MR in SMCs prevented TAC-induced kidney injury.

## 4. Discussion

Tac is a CI commonly used for preventing allograft rejection during organ transplantation. However, ~30% of patients who undergo transplantation present high blood pressure in the first three months due to the use of CIs, and anti-hypertensive therapy, is provided to avoid organ and tissue damage [34,35,36,37]. Several multicenter and randomized clinical trials carried out on liver or kidney transplant recipients have revealed that approximately 12–40% of individuals treated with Tac exhibit nephrotoxicity one year after transplantation [38,39,40,41]. To determine the mechanisms by which Tac causes these detrimental effects, some empirical studies on animals have shown that Tac administration (5 to 12 mg/Kg/day) causes vasoconstriction of glomerular arterioles, thus, decreasing renal blood flow, reducing the GFR, and promoting endothelial dysfunction [42,43,44].

Here, we found that daily administration of 10 mg/Kg/day Tac (Tac-10) for 10 days was associated with high blood pressure, an increase in plasma creatinine, and renal damage, represented by the induction in IL-6 and NGAL, two mediators linked to the proinflammatory status [45,46,47,48]. The dose of Tac used was several times higher than the dose administered to transplant patients [49,50]. However, a study found that C57BL/6 mice might show differences in the resistance to nephrotoxicity induced by drug models (for example, cisplatin) [51] or to other acute kidney injury (AKI) experimental models driving to chronic kidney disease [52]. The administration of Tac-10 in WT and transgenic mice in this study showed reproducible and reliable effects, similar to those reported in other studies, where Tac-10 [15] or a higher dose of Tac [42] was administered to rodents.

Given the brief duration of Tac administration in our experiments (10 days), we classified the resulting nephrotoxicity as acute rather than chronic. This classification is supported by the presence of isometric vacuolation in proximal tubules, which has been observed in both patients and rodents [53,54]. In addition, the induction of IL-6 and NGAL at the renal level was associated with the nephrotoxicity induced by the administration of Tac-10 and a rise in the blood pressure in mice. High blood pressure is a sub-inflammatory condition mainly affecting the cardiovascular and renal organs [55]. In this context, IL-6 is an important cytokine that polarizes T-helper lymphocytes and leads to the immunological imbalance associated with high blood pressure [56,57]. Regarding NGAL, we found in another study that high levels of NGAL can influence hypertension induced by the activation of MR [58], also involving modulation of the immune system [59]. These findings might be relevant as transplant patients treated with Tac often show an imbalance of T-helper lymphocytes in peripheral blood [60,61,62], which might affect the target organs of hypertension. However, more studies are needed in this context to fully elucidate the effects of Tac on the T-helper population.

Hoorn et al. showed that Tac administration in mice activates the renal sodium-chloride cotransporter (NCC), causing hypertension and promoting an increase in the level of aldosterone in plasma [14]. High levels of aldosterone modulate the activity of the NCC [63,64] and activate the renal MR, driving fibrosis, vasoconstriction, inflammation, and oxidative stress [22,56,65,66,67]. These observations and other findings suggest that MRAs might alleviate CIs-induced nephrotoxicity in kidney transplant patients [21]. The use of Spiro is associated with a protective effect against lowering GFR, proteinuria, and glomerulosclerosis in transplanted patients treated with Tac [20,24,25]. In our experimental model, Spiro (100 mg/Kg/day) effectively mitigated hypertension, plasma creatinine elevation, and renal overexpression of IL-6 and NGAL in Tac-treated mice. Nevertheless, further investigations are required to establish its effectiveness in humans. In light of this, we postulated that nonsteroidal MRAs, such as Finerenone, could serve as potential agents for averting kidney allograft damage and CIs-induced nephrotoxicity, as they can efficiently block MR and have fewer adverse effects [68,69].

We previously found that MR deletion from the endothelium does not affect acute CsA-induced nephrotoxicity [27]. However, suppressing the expression of MR in SMC, modulated the activity of L-type calcium (Ca^2+^) channels and decreased the phosphorylation of essential proteins for SMC contraction. These changes regulated microvascular contraction and renal hemodynamics during the administration of CsA [27,70]. This effect was independent of the phosphorylation of endothelial nitric oxide synthase (eNOS) [27], suggesting that the vascular effect of CIs might be attributed to the activation of MR in SMC instead of the MR expressed in endothelial cells. Although we did not investigate these mechanisms related to the nephrotoxicity induced by Tac, we hypothesized that the production of nitric oxide (NO) performs a key role, considering that Tac decreases NO synthesis and endothelial vasodilator function in mice by negatively altering intracellular Ca^2+^ and eNOS phosphorylation [71]. Additional studies have also found that SMC-MR is required for the NO and Ca^2+^ signaling pathways [72,73] and it also performs a critical role in AKI. For example, Barrera-Chimal et al. showed that the SMC-MR-KO mice were protected from an increase in creatinine levels and the induction of NGAL. They also had a lower percentage of injured tubules compared to control mice that underwent to ischemia-reperfusion model [74]. Conversely, Biwer et al. have recently revealed that SMC-MR is linked to aortic stiffness and microvascular myogenic tone in preeclampsia mice subjected to postpartum hypertensive stimuli [75], suggesting its impact on overall circulation. To recapitulate, our findings offer further insights that can be utilized in future investigations to evaluate the efficacy of MRAs in preventing Tac-induced nephrotoxicity and hemodynamic alterations.

## Figures and Tables

**Figure 1 pharmaceutics-15-01373-f001:**
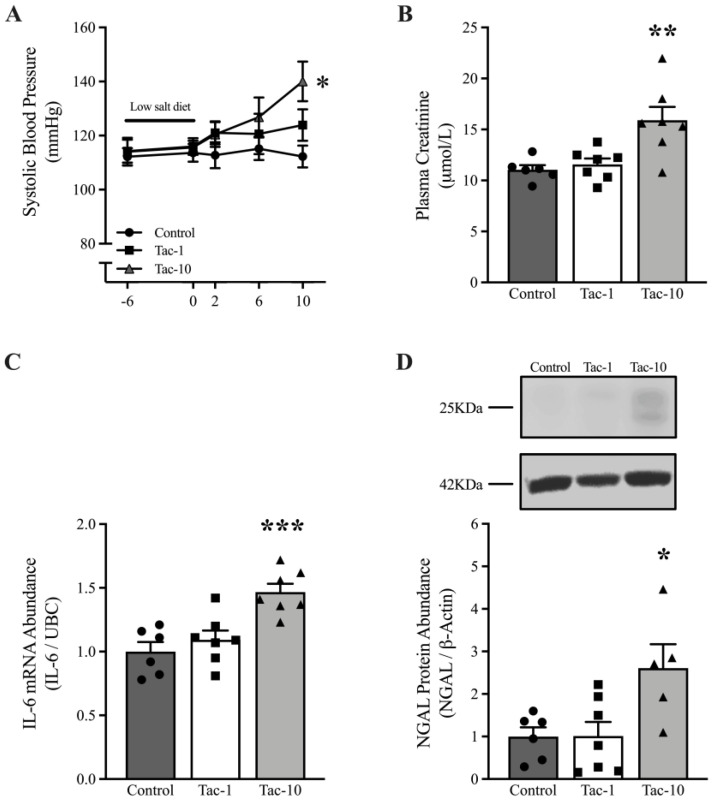
Tacrolimus (10 mg/Kg/d, Tac-10) promotes an increase in blood pressure and renal damage in WT mice after 10 days. Systolic blood pressure was determined by administering low-salt-diet treatment and Tac until day 10 (**A**). Plasma creatinine (**B**), IL-6 mRNA abundance (**C**), and NGAL protein abundance (**D**) at the renal level were determined after 10 days of Tac treatment with doses of 1 mg/Kg/d (Tac-1) and 10 mg/Kg/d (Tac-10). For qRT-PCR and Western blotting analyses, Ubiquitin-C (UBC) and β-Actin were employed as the housekeeping gene and loading control, respectively. A representative image of NGAL and β-Actin expression is shown in the insert D. The data are presented as mean ± SEM (*n* = 3–7). One-way ANOVA; * *p* < 0.05, ** *p* < 0.01, and *** *p* < 0.001 vs. Control.

**Figure 2 pharmaceutics-15-01373-f002:**
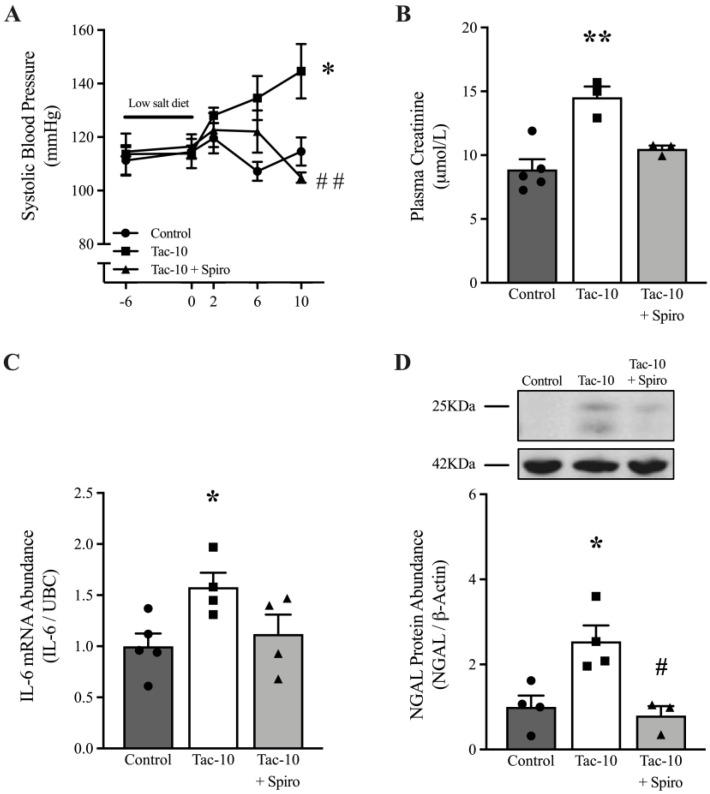
High blood pressure, renal dysfunction, and NGAL overexpression induced by Tac-10 in WT mice were prevented by spironolactone. Systolic blood pressure was determined by administering low-salt-diet treatment along with Tac-10 and 100 mg/Kg/d spironolactone (Spiro) (**A**). Plasma creatinine (**B**), IL-6 mRNA expression (**C**), and NGAL protein abundance (**D**) at the renal level were measured after administering Tac-10 and Spiro for 10 days. UBC and β-Actin were used as the housekeeping gene and loading control for qRT-PCR and Western blotting assays, respectively. A representative image of NGAL and β-Actin expression is shown in the insert D. The data are expressed as the mean ± SEM (*n =* 3–6). One-way ANOVA; * *p* < 0.05 and ** *p* < 0.01 vs. Control; ^#^
*p* < 0.05 and ^##^
*p* < 0.01 vs. Tac-10.

**Figure 3 pharmaceutics-15-01373-f003:**
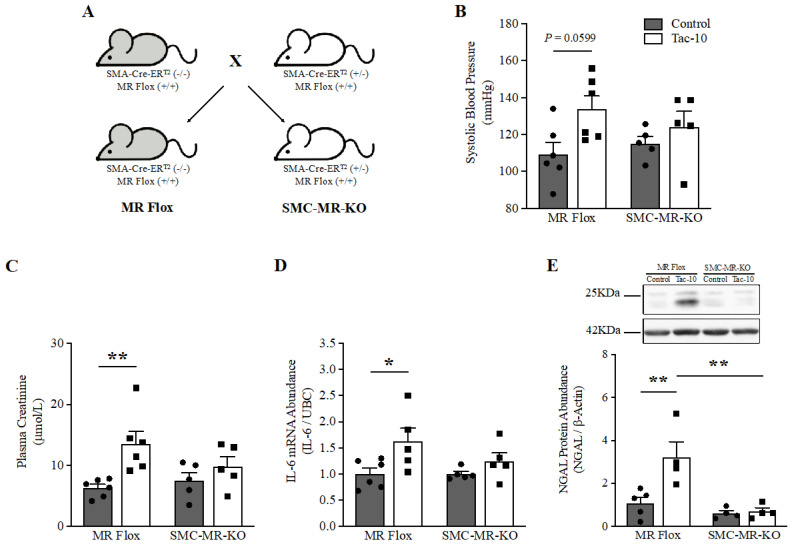
The deletion of MR in smooth muscle cells prevents renal injury induced by tacrolimus. An overview of the Cre-loxP system of deletion of MR in the SMA promoter (**A**). The systolic blood pressure was determined by administering low-salt-diet treatment along with Tac-10 (**B**). Plasma creatinine (**C**), IL-6 mRNA expression (**D**), and NGAL protein abundance (**E**) in the kidneys were determined after 10 days of treatment with Tac-10 in MR Flox and SMC-MR-KO mice. UBC and β-Actin were used as the housekeeping gene and loading control for qRT-PCR and Western blotting assays, respectively. A representative image of NGAL and β-Actin expression is shown in the insert E. The data are expressed as the mean ± SEM (*n* = 4–6). Two-way ANOVA was performed; * *p* < 0.05 and ** *p* < 0.01. The dark gray and white bars represent Control and Tac-10, respectively.

## Data Availability

Not applicable.

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
