# Peer review of "The Mineralocorticoid Receptor on Smooth Muscle Cells Promotes Tacrolimus-Induced Renal Injury in Mice"

_pharmaceutics, 2023, doi:10.3390/pharmaceutics15051373_

Round 1

Reviewer 1 Report

The authors aim to study the role of the mineralocorticoid receptor n the renal damage caused by Tacrolimus, a calcineurin inhibitor commonly used as immunosuppressor following organ transplantation. They used wildtype mice treated with an MR antagonist or mice with a targeted deletion of MR in smooth muscle cells (SMC-MR-KO) and their littermate controls following 10days of tacrolimus treatment. The authors could demonstrate that an increase in blood pressure and plasma creatinine, the renal induction of the interleukin (IL)-6 mRNA expression and the overexpression of the neutrophil gelatinase-associated lipocalin (NGAS), a marker of tubular damage were blunted in mice treated with MR antagonist or in SMC-MR-KO mice.

The topic is highly relevant, although not completely new. The same approach has been published previously testing the immunosuppressive drug cyclosporin A suspected to cause nephrotoxicity via the mineralocorticoid receptor (Amador et al., 2016, “Deletion of the mineralocorticoid receptors in smooth muscle cells blunts renal vascular resistance following acute cyclosporine administration”). Two of the present authors were co-authors on the former paper. Overall, the experiments are well performed, although for some groups (not further indicated) only n=3 mice were analyzed. 

Criticisms: 

1)     Apart from the role of the MR in the tacrolimus-induced nephrotoxicity, the study is lacking mechanistic analysis; is it the same mecanism as in the Amador et al., 2016 paper ? Did the authors look whether there is a decrease of regulatory T lympocytes cells linked to hypertension as described (Chiasson et al., 2011). Are L-type Ca2+  (Cav1.2) involved in the MR-mediated mechanisms as described in Amador et al., 20’16; is the nitric oxide synthase function and/or eNOS phosphorylation reduced as described in Cook et al., 2009; does Tacrolimus administration activates the renal sodium-chloride cotransporter (NCC) causing hypertension and promoting and aldosterone rise in the plasma as described by Hoorn et al., 2011 ?; is the endothelin B modification implicated in this mechanism as described by Barrera-Chimal et al., 2016. The analysis of one or more of those targets would be helpful here.

2)     The description of the figures should be improved; all figures should be presented as dot blots to see the variation amongst the individuals, especially since the number of some groups is really low (n=3); bands seen in the Western blots should be better labeled; why do we see 2 bands for the NGAL protein and which one is the relevant? Controls are missing here.

3)     Was the plasma really stored at 20°C as described in the Mat & Meth section ? 

4)     The authors found differences for the systolic blood pressure, what about the diastolic one ?

5)     It should be specified which figure was calculated with which statistical test.

6)     The discussion does not include the present data and is rather an extensive review of already existing literature on tacrolimus and MR-induced nephrotoxicity: therefore, the discussion should be more focused on the data in the paper described.

Author Response

1) Apart from the role of the MR in the tacrolimus-induced nephrotoxicity, the study is lacking mechanistic analysis; is it the same mechanism as in the Amador et al., 2016 paper? Did the authors look whether there is a decrease of regulatory T lymphocytes cells linked to hypertension as described (Chiasson et al., 2011). Are L-type Ca2+  (Cav1.2) involved in the MR-mediated mechanisms as described in Amador et al., 20’16; is the nitric oxide synthase function and/or eNOS phosphorylation reduced as described in Cook et al., 2009; does Tacrolimus administration activates the renal sodium-chloride cotransporter (NCC) causing hypertension and promoting and aldosterone rise in the plasma as described by Hoorn et al., 2011 ?; is the endothelin B modification implicated in this mechanism as described by Barrera-Chimal et al., 2016. The analysis of one or more of those targets would be helpful here.

Answer 1: We thank to the reviewer for addressing these important questions. Unfortunately, in this opportunity was not possible to analyze deeper the mechanisms discussed in our previous article, such as the nitric oxide status in vasculature or the role of calcium channel (doi: 10.1038/ki.2015.312). For these reasons and considering these limitations, we presented the manuscript as a ‘Communication’ more than a full article. Nevertheless, we improved our discussion according to the data presented here, and according to the inquiries of the reviewer.

2) The description of the figures should be improved; all figures should be presented as dot blots to see the variation amongst the individuals, especially since the number of some groups is really low (n=3). Bands seen in the Western blots should be better labeled; why do we see 2 bands for the NGAL protein and which one is the relevant? Controls are missing here.

Answer 2: We thank to the reviewer, and we agree this commentary. We improved all graphs considering columns and dots. In addition, we improved the description of all legends by also specifying the number of animals and the statistical test for each figure.

Concerning the NGAL protein in the Western blot, we labeled the bands in figures as suggested. The two bands correspond to the non-glycosylated ~22-kDa precursor and glycosylated ~25-kDa mature NGAL (also named ‘Lcn2’). These two forms can be separated according the resolutive capacity of the gel and are recognized by the antibody that we used in our laboratory on kidneys (Abcam #ab63929), but also can be recognized by antibodies from other companies. This have been observed by us and others in different studies:

  • DOI: 10.1172/JCI42004, please see figure 1E.
  • DOI: 10.1038/ki.2015.312, please see figure 3B and 3D.
  • DOI: 10.1186/s12964-018-0285-3, please see figure 5E.
  • DOI: 10.3389/fphar.2021.593682, please see figure 5.

3) Was the plasma really stored at 20°C as described in the Mat & Meth section? 

Answer 3: We appreciate reviewer’s careful reviewing. We apologize for this mistake because plasmas were stored at -20ºC. This was corrected in the revised version of the manuscript (in red, line 95).

4) The authors found differences for the systolic blood pressure, what about the diastolic one?

Answer 4: We also recorded the diastolic blood pressure (DBP) at the end of the experiment. The following figures show that Tac 10mg/kg/d also induced an increase of DBP after 10-days of treatment (Figure I, enclosed), which was prevented by Spironolactone (Figure II, enclosed) and by the MR-ablation from smooth muscle cells in mice (Figure III, enclosed). These results are now mentioned in the last version of the manuscript as data not shown. Thank you for this observation.

5) It should be specified which figure was calculated with which statistical test.

Answer 5: We thank to the reviewer, and we agree this commentary. We improved the description of all legends by also specifying the number of animals and the statistical test for each figure.

6)     The discussion does not include the present data and is rather an extensive review of already existing literature on tacrolimus and MR-induced nephrotoxicity: therefore, the discussion should be more focused on the data in the paper described.

Answer 6: We improved the discussion section according to the results here presented, as we mentioned also in answer 1. Thank you for this recommendation.

Reviewer 2 Report

Calcineurin inhibitors (CNI) nephrotoxicity is a major problem in clinical practice. This paper is very interesting because it focuses on mineralocorticoid receptors (MR) in vascular smooth muscle cells as a contributing factor to CNI nephrotoxicity. Since this is a simple experiment, a more detailed discussion is needed.

(1) Because the drug doses are so high, the authors should also describe changes in body weight and food intake.

(2) The observation period of the experiment was very short (10 days), and a limitation should be added that the authors might be observing Tac-induced Acute Kidney Injury (AKI), not Chronic Kidney Disease (CKD) as experienced in actual clinical practice.

Author Response

Calcineurin inhibitors (CNI) nephrotoxicity is a major problem in clinical practice. This paper is very interesting because it focuses on mineralocorticoid receptors (MR) in vascular smooth muscle cells as a contributing factor to CNI nephrotoxicity. Since this is a simple experiment, a more detailed discussion is needed.

Answer: We thank to the reviewer for the observation. We improved our discussion according to the data presented here and according to this appreciation and from other reviewers (highlighted in red in the revised version of the manuscript).

(1) Because the drug doses are so high, the authors should also describe changes in body weight and food intake.

Answer 1: Thank you for the comments. As we show in the enclosed figure, the administration of tacrolimus (both doses) was not associated to a significant reduction in the mice body weight that we standardized by the tibia length at the end of the experiment. This observation, as the feed condition of animals, were included in Material and Methods of the last revised version.

(2) The observation period of the experiment was very short (10 days), and a limitation should be added that the authors might be observing Tac-induced Acute Kidney Injury (AKI), not Chronic Kidney Disease (CKD) as experienced in actual clinical practice.

Answer 2: We thank to the reviewer, and we agree this observation. As part of revised discussions in the last version of the manuscript (highlighted in red), we have mentioned the nephrotoxicity by tacrolimus and the role MR on smooth muscle cells as acute more than chronic. New evidence related to the acute kidney injury have been discussed to strength this point.

Reviewer 3 Report

In the manuscript, the authors describe that The Mineralocorticoid Receptor on Smooth Muscle Cells Promotes the Tacrolimus-Induced Renal Injury in Mice. However, this manuscript does not present sufficient experimental results to prove the conclusion.

1. In this study, Tacrolimus is treated at the doses of 1 or 10 mg/kg, and only 10 mg/kg dose presented renal injury. However, 10 mg/kg of the dose in animals is very high compared to the dose actually applied to humans in clinical practice. Therefore, it is not an experimental setting that can reflect the clinical situation.

2. In Figure 2, the treatment of spironolactone is not specific to the MR of SMA.

3. The authors employed the SMA-MR-KO mice. However, this paper does not provide genetic analysis results for mice. In Figure 3, the asterisk indication for statistical significance is not clear.

4. The mechanism by which MR expressed in SMC mediates kidney injury has not been provided at all.

Author Response

In the manuscript, the authors describe that The Mineralocorticoid Receptor on Smooth Muscle Cells Promotes the Tacrolimus-Induced Renal Injury in Mice. However, this manuscript does not present sufficient experimental results to prove the conclusion.

1) In this study, Tacrolimus is treated at the doses of 1 or 10 mg/kg, and only 10 mg/kg dose presented renal injury. However, 10 mg/kg of the dose in animals is very high compared to the dose actually applied to humans in clinical practice. Therefore, it is not an experimental setting that can reflect the clinical situation.

Answer 1: We thank to the reviewer for this point of view. In fact, the dose of Tacrolimus used in our work was several times higher than the dose administered to transplant patients. However, it has been reported that C57BL/6 mice show differences in the resistance to nephrotoxicity induced by experimental models induced by drugs (for example, cisplatin, doi:10.1038/ki.2015.327), or susceptibility to acute kidney injury experimental models driving to chronic kidney disease (doi:10.1152/ajprenal.00199.2018). The administration of Tacrolimus in dose of 10mg/kg/d in mice showed a reproducible and reliable effects, which is critical for an experimental model. For these reasons, we kept that dose for the studies with spironolactone and in SMC-MR-KO mice.  Moreover, other studies have reported this same dose of Tacrolimus (doi:10.1161/HYPERTENSIONAHA.110.162917) or a higher dose in rodents (doi:10.1038/hr.2014.79). We describe and discuss this in the revised version of the manuscript.

2) In Figure 2, the treatment of spironolactone is not specific to the MR of SMA.

Answer 2: We refer the effects of spironolactone such as a general MR-antagonism. We verified this in the revised version of the manuscript and only in the studies done in SMC-MR-KO mice we refer the antagonism as specific. We consider as important that even when spironolactone generates a broad MR-blockage, its use during Tac model replicated in a very close way the results observed in SMC-MR-KO mice. By this way, we propose in the discussion that new studies by using MR antagonism in transplanted patients will contribute to the knowledge concerning its potential use for the prevention of nephrotoxicity induced by calcineurin antagonists.

3) The authors employed the SMA-MR-KO mice. However, this paper does not provide genetic analysis results for mice. In Figure 3, the asterisk indication for statistical significance is not clear.

Answer 3: We thank to the reviewer for this observation. This transgenic mouse model has been used in previous publications by our group (DOI: 10.1681/ASN.2016040477) (doi: 10.1038/ki.2015.312), and the genetic ablation of MR from SMC has been previously reported as we show also in the enclosed Figure I (doi:10.1161/HYPERTENSIONAHA.113.01967). In all cases, we verified the genotyping of animals before to do the experiments. All these considerations were included in the revised version of the manuscript (highlighted in red).

In the case of Figure 3, we improved the description of legend by also specifying the number of animals and the statistical test (two-way ANOVA). In the case of Figure 3B, we wrote the exact p value, however, and as described in Materials and Methods, the differences were statistically significant at p <0.05.

4) The mechanism by which MR expressed in SMC mediates kidney injury has not been provided at all.

Answer 4:  We thank to the reviewer for addressing this important point. Unfortunately, in this opportunity was not possible to analyze deeper the mechanisms discussed in our previous article (doi: 10.1038/ki.2015.312), such as the nitric oxide status in vasculature or the role of calcium channel. For these reasons and considering these limitations, we presented the manuscript as a ‘Communication’ more than a full article. Nevertheless, we improved our discussion according to the data presented here, and according to the inquiries of the reviewer.

Reviewer 4 Report

This manuscript has investigated the impact of high dose tacrolimus on acute kidney injury in a mouse model examining the impact of mineralocorticoid inhibition with spironolactone and  in mice with targeted deletion of vascular smooth muscle cell (SMC)mineralocorticoid receptors. They suggest that MRA is providing protection via reduction in SMC activity.

With regards to the data presented, I would like to have seen actual histological evidence of injury - location of this? (histology should include H&E as well as Massons or sirus red demonstration of fibrosis). Was it confined to the proximal tubular epithelial cells and where - juxtamedullary location is where it would be expected if it was an ischaemic mediated injury. 

Likewise given that most of the actions of MRA with respect to injury and fibrosis are on non-vascular pathways as opposed to SMC eNaC mediated changes in cellular contractility - more discussion of how protection may be mediated is required. 

Some more specific immunohistochemistry to demonstrate actual cellular changes in inflammatory pathways rather than western blotting that is just analysing whole tissue rather than specific sites of action. Likewise some in-situ hybridisation of IL6 to demonstrate that the changes are in fact related to the vascular SMC would strengthen the manuscript.

There is no discussion of the impact of SMC MR deletion on the rest of the circulation - is there an effect?

The doses used in this model are industrial rather that pharmacologic and hence the relevance to the human situation is limited.

Tacrolimus at 10mg/kg/day would equate to a human dose in an average male of 700mg/day. Likewise the even higher doses of spironolactone 100mg/kg/day equates to a human dose of 7000mg daily

No justification of the dosing schedule has been provided. Given that Tacrolimus at 1mg/kg/day did not produce any evidence of injury - the relevance of the very high dose needs to be justified.

Author Response

1) With regards to the data presented, I would like to have seen actual histological evidence of injury - location of this? (histology should include H&E as well as Massons or sirus red demonstration of fibrosis). Was it confined to the proximal tubular epithelial cells and where - juxtamedullary location is where it would be expected if it was an ischaemic mediated injury.

Answer 1: We thank to the reviewer for this point of view. According to the short protocol of Tacrolimus administration performed in our work (10 days), we considerate the effects as acute more than chronic. This nephrotoxicity includes isometric vacuolation of proximal tubules in patients and rodents (doi:10.1038/ncpneph0225) (DOI 10.1211/jpp.58.11.0015), which also has been published by us in a previous work where we used cyclosporine-A in mice (doi: 10.1038/ki.2015.312). Therefore, we have done this histological analysis by using H&E on renal sections from the experimental set of Tacrolimus dosage in order to address the inquiry of the reviewer.

As we show in the enclosed representative images and quantification (done by two people in a blinded observations), Tacrolimus treatment 10mg/kg/d was associated to a tubular vacuolation in the proximal cells of kidney mice. Unfortunately, the quality of kidneys sections for the experimental set done during the use of Spironolactone and in SMC-MR-KO mice did not allow us to study the staining by H&E. For these reasons, and considering these limitations, we presented the manuscript as a ‘Communication’ more than a full article, however we discuss this point in the revised version of the manuscript.

2) Likewise given that most of the actions of MRA with respect to injury and fibrosis are on non-vascular pathways as opposed to SMC eNaC mediated changes in cellular contractility - more discussion of how protection may be mediated is required. 

Answer 2: We thank to the reviewer for this important comment. We improved the discussion section according to this recommendation and considering a possible mechanism in the new version of the manuscript (highlighted in red).

3) Some more specific immunohistochemistry to demonstrate actual cellular changes in inflammatory pathways rather than western blotting that is just analysing whole tissue rather than specific sites of action. Likewise some in-situ hybridisation of IL6 to demonstrate that the changes are in fact related to the vascular SMC would strengthen the manuscript.

Answer 3: We thank to the reviewer for addressing these important questions. Considering that the lesions induced by Tac were more frequent at tubular level (Answer 1), and according to previous studies where the NGAL mRNA abundance is also increase in proximal tubules during an inflammatory kidney lesion (doi:10.1172/JCI42004), we think that the pro-inflammatory renal lesion reflected by the increase of IL-6 and NGAL are a consequence of the hemodynamics changes during the administration of Tac.

4) There is no discussion of the impact of SMC MR deletion on the rest of the circulation - is there an effect?

Answer 4: We thank to the reviewer for this observation. Regarding the hemodynamics changes during the administration of calcineurin inhibitors, it has been demonstrated that Cyclosporine-A increases the renal vascular resistance which is dependent on MR in SMC (doi: 10.1038/ki.2015.312). Recently, the group led by Dr. Jaffe has reported the role of SMC-MR in preeclampsia suggesting its impact on the rest of circulation, for instance, on microcirculation and on mesenteric circulation (DOI: 10.1161/CIRCRESAHA.122.321228).

In this current paper, we analyzed the arterial blood pressure as a hemodynamic parameter, which was also associated to the role of MR in SMC. Here, we also recorded the diastolic blood pressure (DBP) at the end of the experiment. The enclosed figures show that Tac 10mg/kg/d also induced an increase of DBP after 10-days of treatment (Figure I), which was prevented by Spironolactone (Figure II) and by the MR-ablation from smooth muscle cells in mice (Figure III). These results (as data not shown), and the potential effect of SMC MR on the rest of the circulation, are now mentioned in the last version of the manuscript. Thank you for this observation.

5) The doses used in this model are industrial rather that pharmacologic and hence the relevance to the human situation is limited.Tacrolimus at 10mg/kg/day would equate to a human dose in an average male of 700mg/day. Likewise the even higher doses of spironolactone 100mg/kg/day equates to a human dose of 7000mg daily. No justification of the dosing schedule has been provided. Given that Tacrolimus at 1mg/kg/day did not produce any evidence of injury - the relevance of the very high dose needs to be justified.

Answer 5: We thank to the reviewer for this point of view. In fact, the dose of Tacrolimus used in our work was several times higher than the dose administered to transplant patients. However, it has been reported that C57BL/6 mice show differences in the resistance to nephrotoxicity induced by experimental models induced by drugs (for example, cisplatin, doi:10.1038/ki.2015.327), or susceptibility to acute kidney injury experimental models driving to chronic kidney disease (doi:10.1152/ajprenal.00199.2018). The administration of Tacrolimus in dose of 10mg/kg/d in mice showed a reproducible and reliable effects, which is critical for an experimental model. For these reasons, we kept that dose for the studies with spironolactone and in SMC-MR-KO mice.  Moreover, other studies have reported this same dose of Tacrolimus (doi:10.1161/HYPERTENSIONAHA.110.162917) or a higher dose in rodents (doi:10.1038/hr.2014.79). We describe and discuss this in the revised version of the manuscript.

Round 2

Reviewer 1 Report

Most of the questions raised had now been answered.

Minor comment: it is still not clear what is the control Western blot sample used in Fig. 1D, Fig. 2D and Fig. 3E. The indication whether only male mice or both sexes have been used for the experiments is missing.  

Author Response

  • Minor comment: it is still not clear what is the control Western blot sample used in Fig. 1D, Fig. 2D and Fig. 3E. The indication whether only male mice or both sexes have been used for the experiments is missing.

Answer 1: In the case of Figures 1D and 2D, we used as control samples those proteins from WT mice kidneys treated with vehicle solution (EtOH 95% and Cremophor Sigma). In the case of Figure 3E, we used as control samples those kidneys from MR flox mice kidneys treated with vehicle solution. It is important to note that the basal NGAL abundance in kidney is modest and depends on the antibody used. Please, see the following references:

  • DOI: 10.1172/JCI42004, please see figure 1E.
  • DOI: 10.1038/ki.2015.312, please see figure 3B and 3D.

In our work, the NGAL quantification was normalized by β-Actin as loading control and received a value of 1 for the control group. By this way, the other groups varied with respect to the respective control.

Concerning the inserts of Western blots, we showed representative images for each experiment (Figures 1D, 2D and 3E), but only showing one sample of each group. In the last version of our manuscript, we comment that the insert for each figure is a representative image of the Western Blot (highlighted in red).

Answer 2: Concerning the sex of mice used in this work, this was indicated in the first line of point 2.2 (highlighted in red).

Reviewer 4 Report

The authors have addressed most of my concerns where they are able to.

Author Response

  • There are some minor grammatical or English construction errors that should be checked for.

Answer 1: We submitted the manuscript undergo a second revision to check the minor grammatical or English construction errors.